# Mechanisms of active diffusion of vesicular stomatitis virus inclusion bodies and cellular early endosomes in the cytoplasm of mammalian cells

Steven J. Moran[1][¤a], Ryan Oglietti[2][¤b], Kathleen C. Smith[3][¤c], Jed C. Macosko[4], George Holzwarth[4], Douglas S. Lyles[1] *

1 Department of Biochemistry, Wake Forest University School of Medicine, Winston-Salem, North Carolina, United States of America, 2 Department of Biology, Wake Forest University, Winston-Salem, North Carolina, United States of America, 3 Department of Chemistry, Wake Forest University, Winston-Salem, North Carolina, United States of America, 4 Department of Physics, Wake Forest University, Winston-Salem, North Carolina, United States of America

¤a Current address: Department of Microbiology & Immunology, Cornell University College of Veterinary Medicine, Ithaca, New York, United States of America
¤b Current address: Wake Forest University School of Medicine, Winston-Salem, North Carolina, United States of America
¤c Current address: National Institute on Aging, Baltimore, Maryland, United States of America
* dlyles@wakehealth.edu

**Data Availability Statement:** All relevant data are within the manuscript and its Supporting Information files.

## Abstract

Viral and cellular particles too large to freely diffuse have two different types of mobility in the eukaryotic cell cytoplasm: directed motion mediated by motor proteins moving along cytoskeletal elements with the particle as its load, and motion in random directions mediated by motor proteins interconnecting cytoskeletal elements. The latter motion is referred to as "active diffusion." Mechanisms of directed motion have been extensively studied compared to mechanisms of active diffusion, despite the observation that active diffusion is more common for many viral and cellular particles. Our previous research showed that active diffusion of vesicular stomatitis virus (VSV) ribonucleoproteins (RNPs) in the cytoplasm consists of hopping between traps and that actin filaments and myosin II motors are components of the hop-trap mechanism. This raises the question whether similar mechanisms mediate random motion of larger particles with different physical and biological properties. Live-cell fluorescence imaging and a variational Bayesian analysis used in pattern recognition and machine learning were used to determine the molecular mechanisms of random motion of VSV inclusion bodies and cellular early endosomes. VSV inclusion bodies are membraneless cellular compartments that are the major sites of viral RNA synthesis, and early endosomes are representative of cellular membrane-bound organelles. Like VSV RNPs, inclusion bodies and early endosomes moved from one trapped state to another, but the distance between states was inconsistent with hopping between traps, indicating that the apparent state-to-state movement is mediated by trap movement. Like VSV RNPs, treatment with the actin filament depolymerizing inhibitor latrunculin A increased VSV inclusion body mobility by increasing the size of the traps. In contrast neither treatment with latrunculin A nor depolymerization of

**Funding:** This study was supported by a grant from the U.S. National Institute of Allergy and Infectious Diseases R01 AI20623 (D.S.L). S.J.M. was supported in part by the National Institutes of Health Training Grant T32 AI007401. We also acknowledge the support of the Cellular Imaging Shared Resource and the Cell Engineering Shared Resource of the Comprehensive Cancer Center of Wake Forest University, which are supported by National Cancer Institute grant P30 CA012197. The funders had no role in study design, data collection and analysis, decision to publish, or preparation of the manuscript.

**Competing interests:** The authors have declared that no competing interests exist.

microtubules by nocodazole treatment affected the size of traps that confine early endosome mobility, indicating that intermediate filaments are likely major trap components for these cellular organelles.

## Introduction

Diffusion (i.e. Brownian motion) is the random motion of small particles caused by the thermal motion of molecules in a fluid. Thermal motion within a cell allows small particles to diffuse freely, whereas the dense network of the cytoskeleton and other cellular structures prevents free diffusion of large particles [1, 2]. The exclusion limit for diffusion in the cytoplasm of mammalian cells is a hydrodynamic diameter of approximately 100 nm [3]. The network of actin filaments is often considered the major barrier to particle diffusion in the cytoplasm [3], although other cellular elements certainly contribute. Distances between cytoplasmic actin filaments visualized by electron microscopy appear to range from around 50 nm in actin-rich areas to >200 nm in areas less actin-rich [4]. Particles too large to freely diffuse in a cell undergo two different types of motion: directed motion mediated by motor proteins moving their cargo along cytoskeletal elements, and motion in random directions referred to as active diffusion. Active diffusion is an energy-dependent process that occurs through motor proteins acting on the cytoskeleton [1, 2]. Motor proteins are primarily known for mediating the directed motion of intracellular cargo, but they also generate effectively diffusive movements through sequences of active movements that are random in direction [1, 5]. Active diffusion is a fundamental cell biological process that allows larger particles to migrate through the cytoplasm, and encounter other structures related to their functions, including the actin filaments and microtubules responsible for directed motion.

Mechanisms of directed motion have been extensively studied and have aided in the conceptualization of particle movement (e.g., [6]). By contrast, active diffusion has been described by more nebulous terminology, such as "anomalous diffusion." Cell-free studies of actin-microtubule composites containing a limited number of components have done much to clarify the competing roles of crowding, spatial heterogeneity, and motor proteins to restrictions on particle mobility and active diffusion [7–9]. The purpose of the experiments described here was to determine how the principles of active diffusion apply to cytoplasmic particles with different physical and biological properties. The results indicate that the principles are similar, but the biology of the interaction with the cytoskeleton differs among different particle types.

We have previously investigated the intracellular movement of the ribonucleoprotein (RNP) core of vesicular stomatitis virus (VSV) as a molecular probe for the mechanism of active diffusion [10, 11]. The VSV RNP core is a loosely-coiled, flexible structure composed of the single-stranded 11-kb VSV RNA genome associated with approximately 1200 copies of nucleocapsid (N) protein, 460 copies of phosphoprotein (P), and 50 copies of large (L) protein. The total mass of the RNP is 87.1 MDa [12, 13]. The RNP is critical for processes like primary and secondary transcription, genome replication, and virus assembly [14]. The hydrodynamic diameter of the RNP is approximately 170 nm [13]. Thus, the RNP is too large to diffuse freely in the cytoplasm of infected cells. It was determined through rapid live-cell fluorescence imaging and variational Bayesian analysis methods that VSV RNPs primarily undergo random motion, and that random motion can be further sub-divided into two categories: (i) confined motion in which the particle bounces back and forth within an area in the cytoplasm and (ii) hopping-like motion in which the particle hops from one area in the cytoplasm to the next

[10]. It was determined that VSV RNPs are trapped in part by actin filaments [11]. Further-more, RNP mobility is enhanced by expansion and contraction of the actin filament traps through the action of non-muscle myosin II, which indicates that VSV RNPs move by active diffusion [11]. The purpose of the experiments presented in this study was to determine whether actin filaments or microtubules compose traps that mediate the random movement of larger particles like VSV inclusion bodies and cellular early endosomes that have physical and biological properties different from those of VSV RNPs.

VSV inclusion bodies are cellular compartments composed of viral RNPs and perhaps solu-ble viral proteins and RNA, that are the major sites of viral RNA synthesis [15, 16]. They are similar to other cellular membraneless compartments formed by combinations of site-specific interactions and/or phase separation [17]. VSV inclusion bodies have properties similar to other membraneless compartments [16] and are typical of inclusion bodies formed by other viruses with negative-sense RNA genomes [15].

Cellular early endosomes are typical cellular membrane-bound organelles that participate in recycling and degradation of macromolecular cargo within the endocytic pathway [18–20]. Directed motion of early endosomes occurs along microtubules and actin filaments [21–23]. However, a substantial amount of their mobility is random in direction and is presumably due to active diffusion.

Like our previous studies [10, 11], the approach for this study used rapid live-cell fluores-cence imaging and variational Bayesian analysis methods to track and analyze particle move-ment. The general pattern of movement of both VSV inclusion bodies and early endosomes was similar to that of VSV RNPs, consisting of periods of confined motion, as though confined in a trap, followed by small hops to a different location, where confined motion continued. Whereas individual VSV RNPs appear to hop from one trap to another [10, 11], the data indi-cate that movement of these larger particles from one area of the cytoplasm to another was due to the dynamic movement of the fibers which make up the trap.

The roles of actin filaments and microtubules in the motion of these particles were deter-mined through treatment with two cellular inhibitors: latrunculin A to depolymerize actin fila-ments [24] and nocodazole to depolymerize microtubules [25]. Like individual VSV RNPs [11], actin filament depolymerization through latrunculin A treatment increased the size of VSV inclusion body traps, indicating that actin filaments are at least part of the mechanism confining the motion of inclusion bodies. Most of the motion of early endosomes was similar to that of VSV inclusion bodies, consisting of confined motion in one area of the cytoplasm with occasional hopping to another area, although there were more instances of directed motion among early endosomes compared to VSV inclusion bodies. In the case of early endo-somes experiencing trapped states, neither actin filament depolymerization through latruncu-lin A treatment nor microtubule depolymerization through nocodazole treatment had a significant effect on the width of trapped states of early endosomes, suggesting that other cellu-lar elements, likely intermediate filaments, are responsible for confining their motion. The results from this study indicate that the basic pattern of active diffusion of each of the particles analyzed consists of bouncing back and forth within a trap with movement from one area of the cytoplasm to another in random directions. However, the biological details of how these particles of different types interact with the cytoskeleton differ.

## Results

### Identification of VSV inclusion bodies in living cells

VSV RNPs and inclusion bodies can be visualized in living cells infected with recombinant viruses that express P protein fused to enhanced green fluorescent protein (eGFP) [16, 26, 27].

Between 3 and 4 hours post-infection (hpi), nearly all of the RNPs are in the form of individual RNPs in lung epithelial (A549) cells infected with recombinant VSV containing eGFP inserted into the hinge region of the P protein (VSV-PeGFP) [10, 11]. VSV inclusion bodies begin to form around 4 hpi [15], and their size increases as infection progresses [27]; however, individual VSV RNPs are also present. To analyze the motion of inclusion bodies, infected cells were imaged by fluorescence microscopy between 4 and 5.5 hpi at 100 frames/s for a total of 400 frames. To differentiate VSV inclusion bodies from individual VSV RNPs, the minimum fluorescence intensity threshold of the images was increased to "blanket-out" nearly all individual RNPs (compare Fig 1A and 1B). The remaining fluorescent particles were determined to be VSV inclusion bodies based on their higher fluorescence intensity above the minimum fluorescence intensity threshold and by their larger size (compare Fig 1C to 1D). The sizes of the inclusion bodies analyzed in these experiments were determined from the area of fluorescence intensity above the threshold and are shown in Table 1 as an approximate diameter calculated as if they were circular. A histogram of the distribution of inclusion body sizes is shown in Fig 1E.

## VSV inclusion body mobility increases following latrunculin A treatment

The purpose of the experiments in Figs 2 and 3 was to determine whether the random motion of VSV inclusion bodies and the effects of actin filament depolymerization through latrunculin A treatment were similar to the effects observed with VSV RNPs [11]. A549 cells were infected with VSV-PeGFP and were then treated or not treated with 0.25 μM latrunculin A for 15 to 30 min between 4.5 and 5.5 hpi. This time period and concentration of latrunculin A depolymerizes 40–60% of filamentous actin without disrupting the flattened morphology of A549 cells [11]. Treatment with latrunculin A under these conditions did not significantly affect the sizes of the inclusion bodies (Fig 1E, apparent diameter 1960 ± 350 nm, n = 113, p > 0.05 compared to controls in Table 1). The tracks of individual VSV inclusion bodies were determined at sub-pixel resolution using Video Spot Tracker (VST) software. Fig 2A shows a typical track of a VSV inclusion body in a control cell that is representative of inclusion bodies analyzed. The data shown are the x and y coordinates of the center of the particle centered around the mean x and y. Like previous experiments with VSV RNPs, VSV inclusion bodies bounced back and forth within the cytoplasm and moved to a different area one or more times per second. However, their bouncing back and forth was less dramatic than that observed with individual VSV RNPs [compare with Fig 2A in [11]]. Furthermore, instead of rapid hops from trap to trap like VSV RNPs, e.g., within one frame [10, 11], VSV inclusion bodies often meandered more slowly from one area in the cytoplasm to the next over several frames. When infected cells were treated with latrunculin A and imaged between 4.5 and 5.5 hpi, there was a similar pattern of bouncing back and forth in one area followed by movement to another area of the cytoplasm (Fig 2B). Analysis of mean squared displacement (MSD) as a function of time delay (τ) for all VSV inclusion body tracks from three independent experiments showed that VSV inclusion body mobility increased slightly upon actin filament depolymerization through latrunculin A treatment (Fig 2C), indicating that actin filaments likely play a role in reducing their mobility.

A variational Bayesian analysis approach used in pattern recognition and machine learning [28] was used to further analyze the motion of VSV inclusion bodies in control and treated cells. This approach models the positions of the particle as two-dimensional (2D) Gaussian distributions of data points and identifies the most probable number of Gaussian states and the properties of those states for particles being analyzed as previously detailed [10, 11, 29]. The results of the variational Bayesian analysis of the representative VSV inclusion body in a

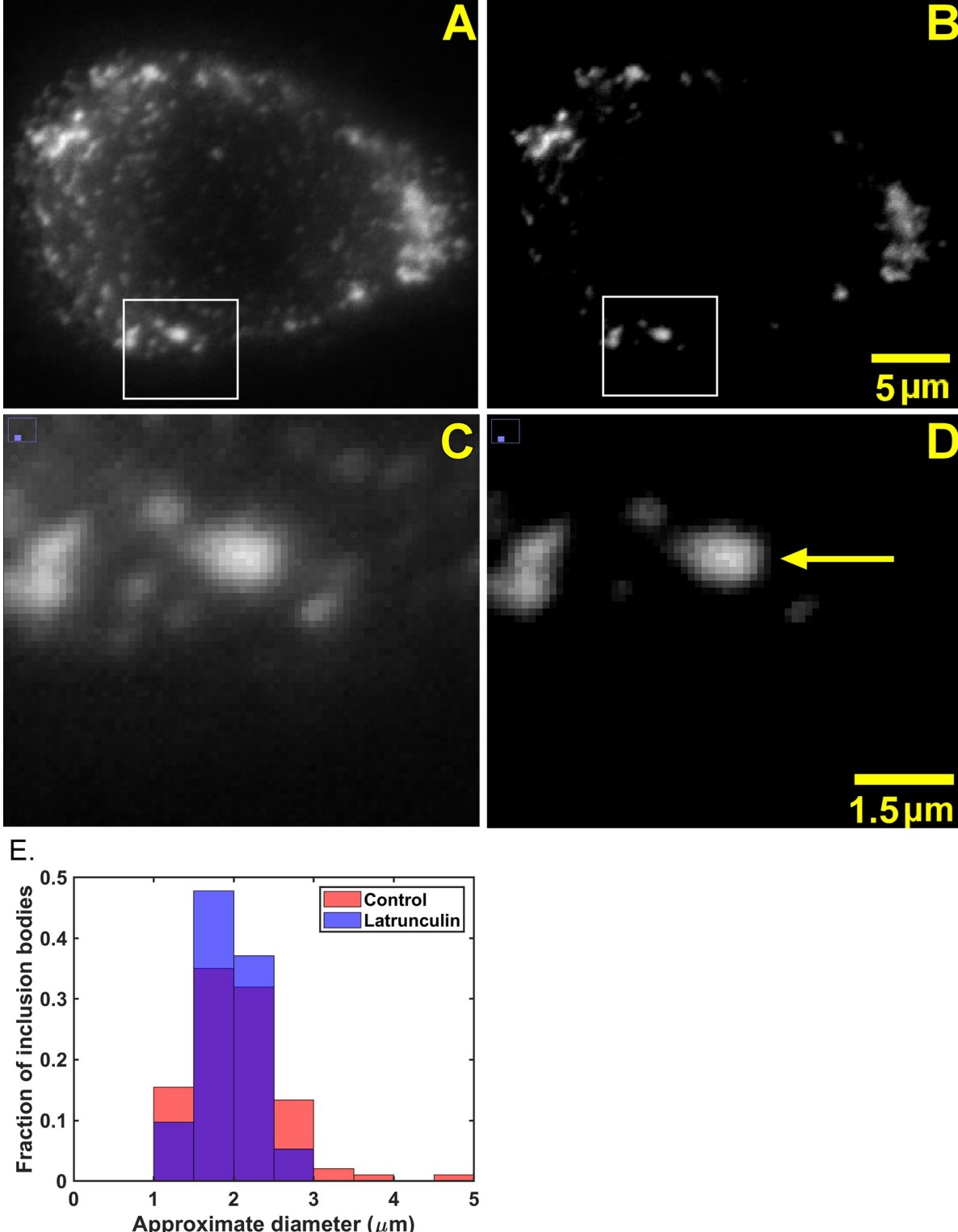

**Fig 1. Distinguishing VSV inclusion bodies from individual VSV RNPs in living cells.** A549 cells were infected with VSV-PeGFP (MOI = 3) and imaged by fluorescence microscopy at 100 frames/s between 4 and 5.5 hpi, a single frame of which is shown (A). (B) The same image as panel A after increasing the minimum fluorescence intensity threshold above that of individual VSV RNPs. (C) Magnified image of panel A. (D) Magnified image of panel B. Arrow indicates example of VSV inclusion body chosen for analysis. (E) Approximate diameters of fluorescently labeled VSV inclusion bodies (n = 97) in control cells and in cells treated with 0.25 μM latrunculin A (n = 113)

were determined by quantifying pixel fluorescence intensities using ImageJ as described in Materials and Methods. The size distributions of the two inclusion body populations were not significantly different (2.03 ± 0.59 μm in controls versus 1.96 ± 0.35 in treated, p > 0.05 by Student's t-test).

control cell (Fig 2A) and a latrunculin A-treated cell (Fig 2B) are shown in Fig 3A and 3B respectively as Bayes plots, in which the position of the center of the particle is color-coded as its most probable state. Each state is represented as an ellipse that is ± two standard deviations in the 2D Gaussian distributions of data points. Fig 3C and 3D show the time dependence of the states occupied by the VSV inclusion bodies in a control cell and treated cell in Fig 3A and 3B, respectively. The VSV inclusion bodies occupied each state for periods ranging from about 100 frames (1 s) to less than 30 frames (0.3 s). Some of the apparent changes in state of short duration were due to overlap between states, in which data points were assigned to one state or the other with highest probability even though their probabilities were similar [10].

The sizes of the states occupied by VSV inclusion bodies were estimated as the averages of the long and short axes of the ellipses in data like those in Fig 3A and 3B. Fig 4A shows a histogram of state widths derived from the VSV inclusion body tracks from the three independent experiments from Fig 2C. The data do not follow a normal distribution, but the distributions were determined to be significantly different (p < 0.05, Mann-Whitney test, median state width = 0.08 μm in control and 0.09 μm in latrunculin A-treated cells). This result likely indicates that actin filaments are part of the traps that confine VSV inclusion body mobility, similar to the results obtained for individual VSV RNPs in latrunculin A-treated cells [11]. Although latrunculin A treatment did not significantly affect the size of the inclusion bodies (Fig 1E), it is also possible that it altered their composition or other physical properties to enhance their mobility.

Data like those in Fig 3C and 3D were used to determine how long VSV inclusion bodies occupied a particular state (dwell time). Dwell times were calculated as the number of frames between transitions from state to state and are shown as a histogram in Fig 4B. The distributions of dwell times were not significantly different in control versus treated cells, indicating that latrunculin A treatment did not significantly affect the rate of movement from state to state, which was also similar to results obtained for individual VSV RNPs.

The average diameter of the VSV inclusion bodies analyzed was approximately 2 μm (Table 1). The data obtained in the Video Spot Tracker (VST) analysis represent the movement of the center of the fluorescent particle. The data in Fig 4A show that the extent of VSV inclusion body motion within each state is much less than the size of the inclusion body itself. It is likely that for larger particles like VSV inclusion bodies, the size of the trap that confines the particle is the diameter of the particle plus the extent of particle movement (i.e. size of the

**Table 1. Sizes of VSV RNPs, VSV inclusion bodies, and cellular early endosomes.**

| Particle | Particle Type | Diameter (nm) |
|---|---|---|
| VSV RNP | Loosely coiled, flexible RNP containing viral genome | 172[a] |
| VSV inclusion body | Membraneless cellular compartment | 2030 ± 590 |
| Early endosome | Cellular membrane-bound organelle | 642 ± 210 |

Approximate diameters of fluorescently labeled VSV inclusion bodies (n = 97) and early endosomes (n = 200) were determined by quantifying pixel fluorescence intensities using ImageJ as described in Materials and Methods. Data are means ± standard deviations.

[a]Hydrodynamic diameter determined by dynamic light scattering [13].

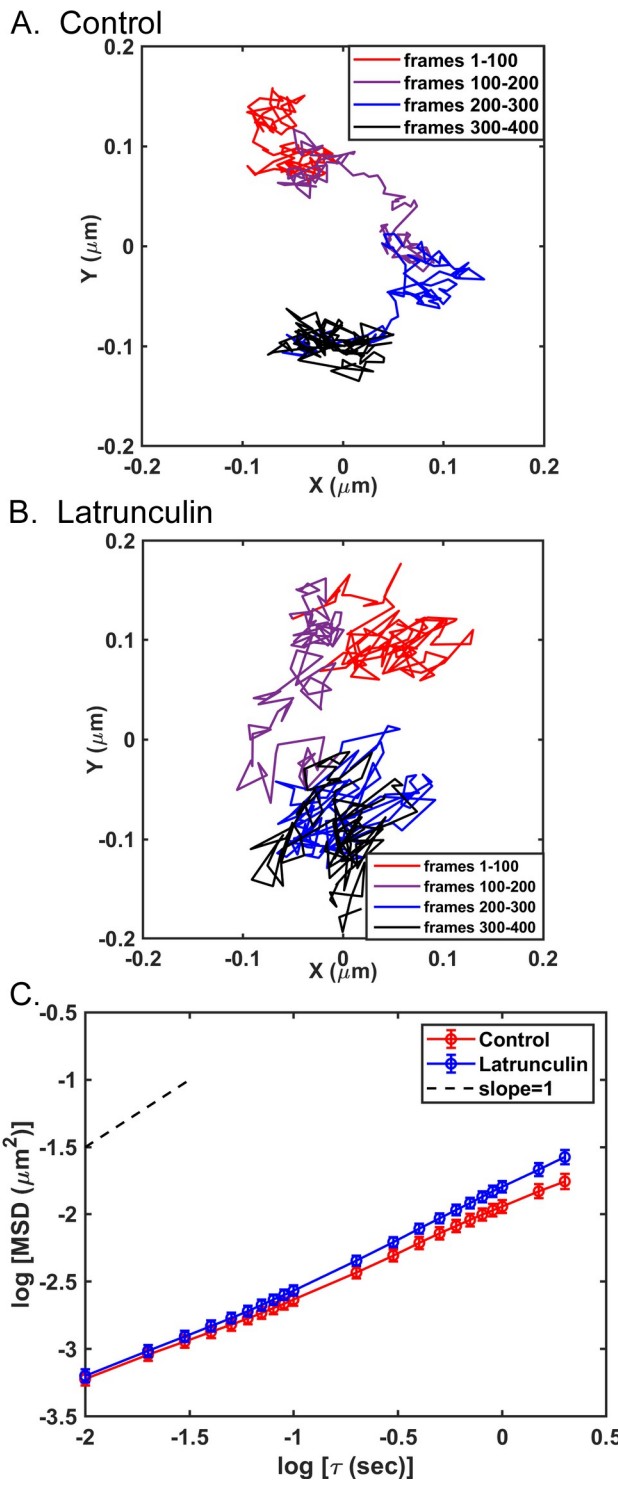

**Fig 2. VSV inclusion body movement in control cells and cells treated with latrunculin A.** A549 cells were infected with VSV-PeGFP (MOI = 3) for 4 hours, then not treated or treated with 0.25 µM latrunculin A. The cells were imaged by fluorescence microscopy at 100 frames/s for 400 frames 15 to 30 min after addition of latrunculin A. Tracks of individual VSV inclusion bodies were analyzed at subpixel resolution with Video Spot Tracker software. Representative tracks of an individual VSV inclusion body from a control (A) or latrunculin A-treated (B) cell are shown centered around their mean $x$ and $y$ positions. The color-coded legend indicates VSV inclusion body movement at each of the 1-s (100-frame) intervals. Mean squared displacements (MSDs) were determined as a function of time delay ($\tau$) and are shown as a log-log plot (C). The data are means ± 95% confidence intervals (n = 188

tracks from control cells and 192 tracks from latrunculin A-treated cells) obtained in three independent experiments. The slopes of the curves are < 1, as seen by comparison to the dashed line (slope = 1.0).

calculated Gaussian state). Further insight into the motion of larger particles comes from considering the distances between states. Fig 4C shows a histogram of the distances from the center of one state to the center of the next. The median distance between states in control cells was 0.07 μm, which increased to 0.09 μm in latrunculin A-treated cells. Although these distances are small relative to the size of the inclusion bodies, they are well above the localization error within the positions of similarly sized fluorescent latex beads stuck to the surface of a cover slip (0.023 μm). If the size of the trap that confines the particle is the diameter of the particle plus the size of the calculated Gaussian state (i.e., >2 μm), the distances between states are much shorter than the size of the traps. Thus, the transition between states cannot be due to

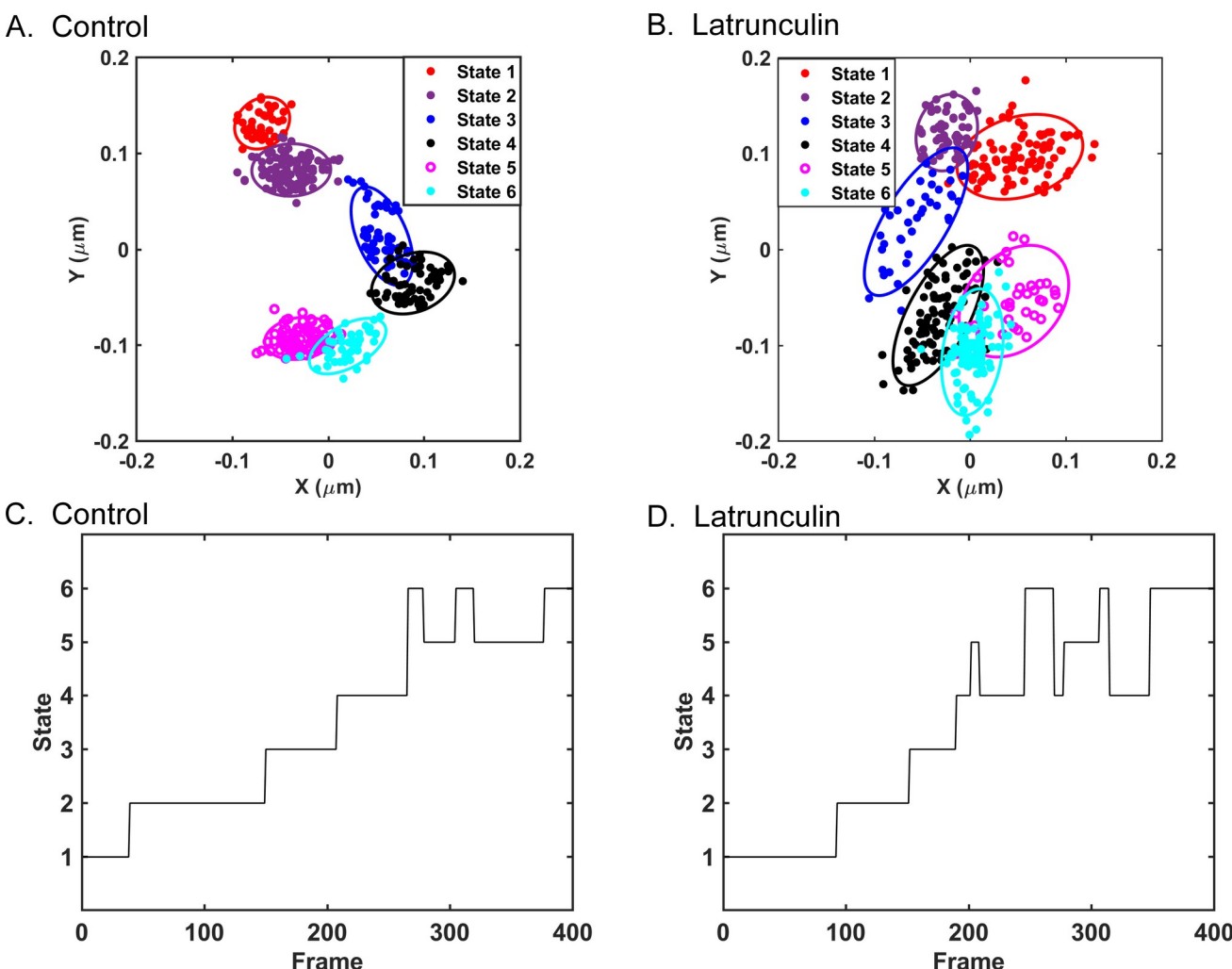

**Fig 3. Variational Bayesian analysis of VSV inclusion body tracks in control cells and cells treated with latrunculin A.** The data shown in Fig 2 were analyzed using a variational Bayesian approach to identify the most probable number of states occupied by individual VSV inclusion bodies and the properties of those states. (A, B) The same data as in Fig 2A and 2B, respectively, for an individual VSV inclusion body from a control (A) or latrunculin A-treated (B) cell represented as a mixture of 2D Gaussian distributions of data points, with ellipses indicating ± two standard deviations in the distributions. (C, D) Time dependence (frame number) of inclusion body occupation of the states identified in panels A and B, respectively.

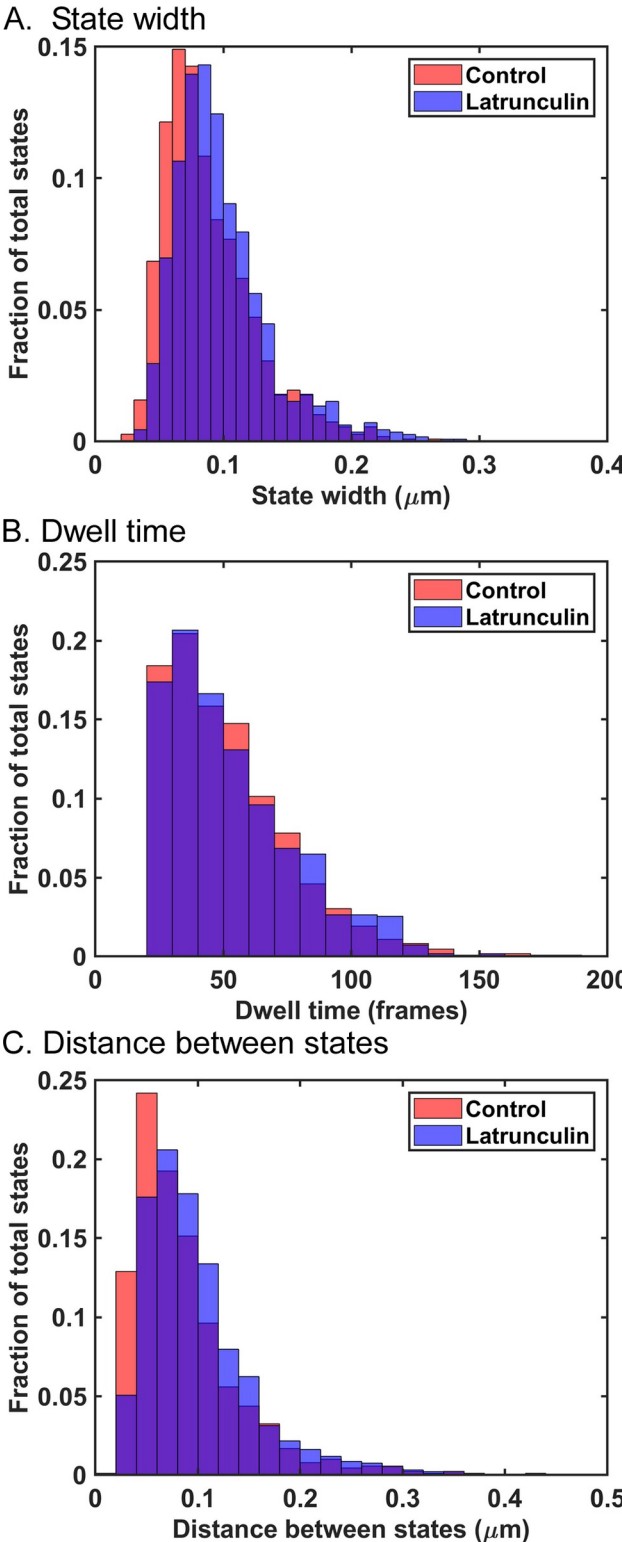

**Fig 4. State widths, dwell times, and distance between states of VSV inclusion body tracks in control cells and cells treated with latrunculin A.** (A) State widths for individual inclusion bodies in control and latrunculin A-treated cells in Fig 2C were calculated as the averages of the two axes in ellipses similar to those in Fig 3A and 3B, and are shown as histograms. States in latrunculin A-treated cells (n = 1118) were significantly larger than those in control cells (n = 1080) (p < 0.05, Mann-Whitney test). (B) Inclusion body dwell times within states were calculated as the time

(number of frames) from one hop between states to the next in data similar to those in Fig 3C and 3D and are shown as histograms. There was no significant difference between dwell times of inclusion bodies in control cells (n = 1085 dwell times) versus latrunculin A-treated cells (n = 1093 dwell times) (p > 0.05, Mann-Whitney test). (C) The distances between states for individual inclusion bodies in control and latrunculin A-treated cells were calculated as the distance from the center of one state to the center of the next and are shown as histograms. State-to-state distances of inclusion bodies in latrunculin A-treated cells (n = 927 distances) were significantly larger than those in control cells (n = 893 distances) (p < 0.05, Mann-Whitney test).

hopping of the inclusion body from one trap to another trap as in the case with individual VSV RNPs. These data suggest that movement of these larger particles from one area of the cytoplasm to another was due to the dynamic movement of the fibers which make up the trap. This hypothesis was supported by analysis of the motion of cellular early endosomes.

## Distinguishing trapped states of early endosomes from those undergoing directed motion

To analyze the motion of early endosomes, A549 cells were transduced with a baculovirus vector that expresses the early endosome marker Rab5a fused to red fluorescent protein (RFP). Transduced cells were imaged at 100 frames/s for 400 frames like VSV inclusion bodies. VSV RNPs and inclusion bodies rarely undergo directed motion [[10, 11]; data not shown]. In contrast, early endosomes frequently underwent directed motion, although random motion was more common. Results of variational Bayesian analysis were used to distinguish between trapped and directed states for early endosomes, as described in [10] and illustrated in Fig 5. Fig 5A shows a segment of a particle track in which the endosome was undergoing directed motion (rotated to be horizontal) and the ellipse resulting from the variational Bayesian analysis. The eccentricity of the ellipse is defined as the ratio of the long axis (± 2 σ major) divided by the short axis (± 2 σ minor). Three criteria were used to classify a state as directed [10]: (i) eccentricity value greater than 2.5, (ii) average slope of log-log plot of MSD vs. τ greater than 0.5, and (iii) the particle starting at one end of the ellipse and traveling mostly in a unidirectional fashion to the other end [i.e. from "point A to point B" as shown in Fig 5A [10]]. In more biological terms, the long axis of the ellipse was interpreted as the distance traveled by a particle occupying a directed state, and the short axis as the extent of random motion of a particle in a directed state (Fig 5B).

Fig 6A shows the track of an individual early endosome occupying only trapped states, and Fig 6B shows the track of another occupying both trapped and directed states in control cells. Bayes plots from the analysis of data from Fig 6A and 6B are shown in Fig 6C and 6D, respectively. Early endosomes occupying trapped states bounced back and forth within a confined area and moved from one area to another (Fig 6A and 6C) like VSV RNPs [10, 11]. Early endosomes undergoing directed states traveled in a unidirectional fashion with little bouncing back and forth within those states, but also experienced periods of time in trapped states (Fig 6B and 6D).

The effects of actin filament or microtubule depolymerization on early endosomes occupying directed states were analyzed by treating transduced A549 cells with either 0.25 μM latrunculin A or 15–20 μM nocodazole, and imaging them between 15 and 30 min after cellular inhibitor treatment. Fig 6E shows the fraction of states of early endosomes classified as directed states in control cells relative to latrunculin A- and nocodazole-treated cells, presented as a bar plot. In control cells, 22% of states occupied by early endosomes were directed states, which was decreased to ~10% in latrunculin A-treated cells and ~6.5% in nocodazole-treated cells (Fig 6E). These data are consistent with previous data establishing the dependence of early endosomes on both microtubules and actin filaments for directed motion throughout the

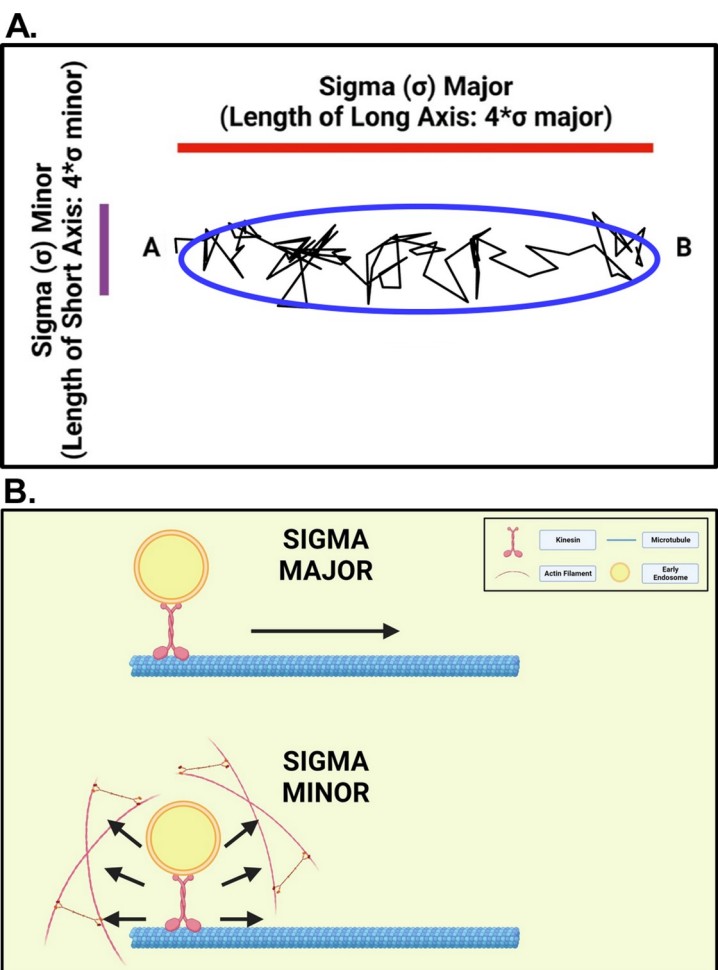

**Fig 5. Interpretation of sigma major and sigma minor in Bayesian analysis of directed states.** (A) Example segment of a particle track in which the endosome was undergoing directed motion (rotated to be horizontal) and the ellipse resulting from the variational Bayesian analysis. The length of the long axis (red line) is 4 x sigma major, i.e., ± two standard deviations in the longest dimension of the 2-D Gaussian distribution of data points. The length of the short axis (purple line) is 4 x sigma minor, the value of the standard deviation in the shortest dimension of the 2-D Gaussian distribution. (B) Diagram showing models of the interpretation of sigma major and sigma minor at the cell biology level, where sigma major reflects the distance traveled by a particle occupying a directed state, and sigma minor reflects the extent of random motion of a particle in a directed state. Figures were created with BioRender software (BioRender.com).

cytoplasm [21–23] and support the assignment of states as either directed or trapped based on the criteria described above.

Fig 7 shows analysis of the mobility of early endosomes that still occupied directed states following the inhibitor treatments used for Fig 6E. It is likely that most of this residual directed motion following treatment with nocodazole was due to motion along actin filaments whereas most of the residual directed motion following treatment with latrunculin A was due to motion along microtubules. Early endosomes occupying directed states had decreased mobility, as measured by their MSD, in nocodazole-treated cells relative to those in control cells (Fig 7A), whereas in latrunculin A-treated cells, early endosomes occupying directed states had increased mobility relative to control (Fig 7B). These data suggest that microtubules are responsible for more of the directed state mobility of early endosomes than actin filaments. The results of the variational Bayesian analysis agreed with the MSD-based results in that the

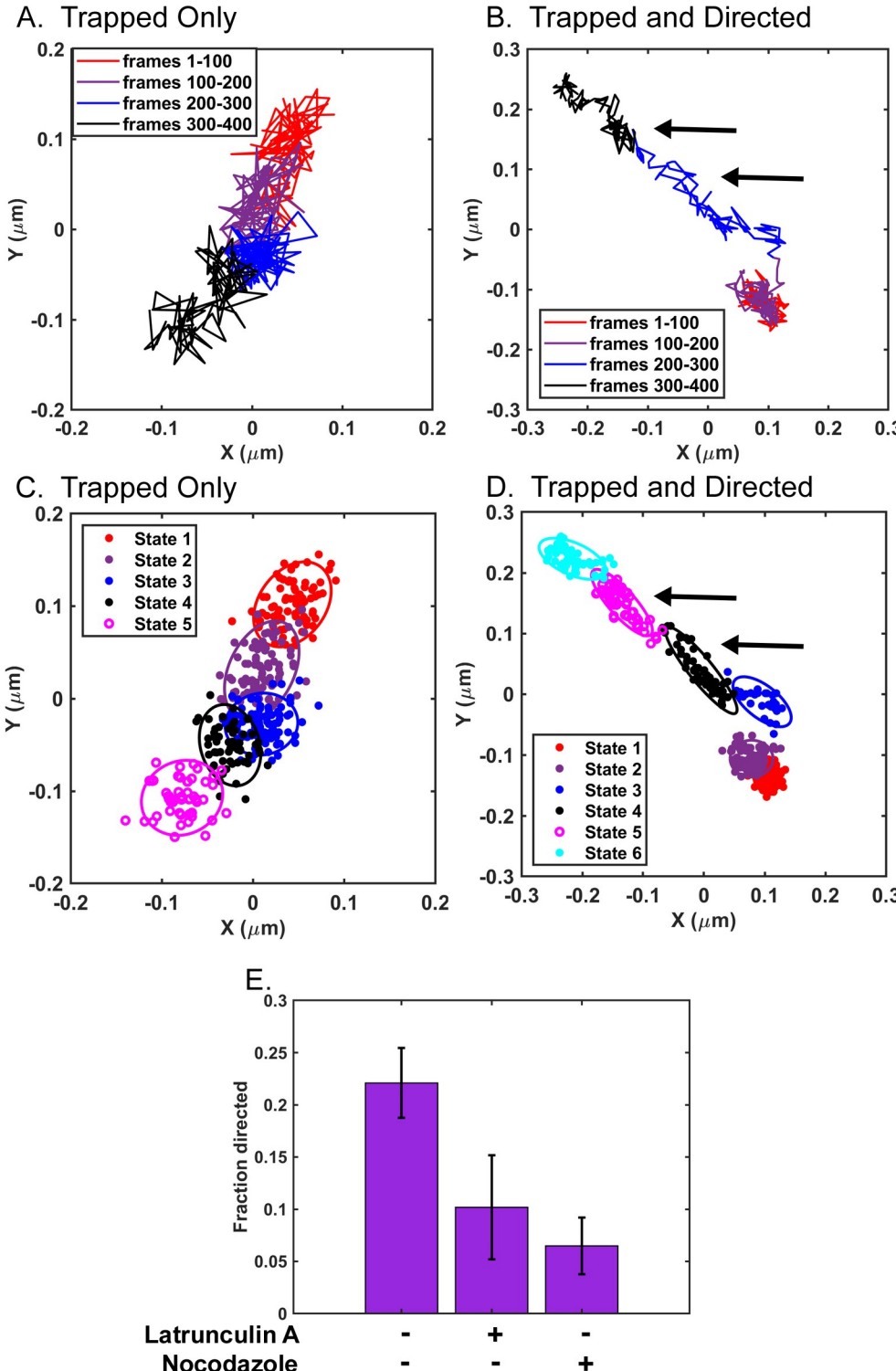

**Fig 6. Movement of early endosomes occupying trapped and directed states.** A549 cells transduced with a baculovirus vector that expresses the early endosome marker Rab5a fused to red fluorescent protein were not treated or treated with 0.25 μM latrunculin A or 15–20 μM nocodazole. Cells were imaged by fluorescence microscopy 15 to 30 min after addition of cellular inhibitor at 100 frames/s for 400 frames. Tracks of individual early endosomes were analyzed at subpixel resolution with Video Spot Tracker software. Representative tracks from an individual early endosome occupying only trapped states (A) or trapped and directed states (B) are shown centered around their mean $x$ and $y$ positions. The color-coded legend indicates early endosome movement at each of the 1-s (100-frame) intervals.

(C, D) Variational Bayesian analysis was used to identify the most probable number of states and the properties of those states for early endosomes in Fig 6A and 6B, respectively. The data are represented as a mixture of 2D Gaussian distributions of data points, with ellipses indicating ± two standard deviations in the distributions. Examples of directed states are indicated by arrows in both Fig 6B and 6D. (E) The fraction of directed states was calculated by dividing the number of directed states by the number of all states analyzed per experiment. Data shown are means ± 95% confidence intervals for n = 6 experiments for control untreated cells and n = 3 experiments for cells treated with latrunculin A or nocodazole.

lengths of the long axes of the ellipses that defined directed states significantly decreased in nocodazole-treated cells (Fig 8A) but increased in latrunculin A-treated cells (Fig 8C). This result is consistent with early endosomes traveling shorter distances on actin filaments than on microtubules. Treatment with nocodazole did not have a significant effect on the lengths of the short axes of the ellipses that defined directed states (Fig 8B), whereas treatment with latrunculin A significantly increased the lengths of the short axes (Fig 8D). This result is consistent with actin filaments playing a role in confining the random motion of early endosomes undergoing directed motion, as illustrated in the model in Fig 5B.

## Role of actin filaments and microtubules in active diffusion of early endosomes

The mobility of early endosomes that did not occupy directed states and only occupied trapped states is likely due solely to active diffusion and was analyzed separately from early endosomes that occupied both directed and trapped states. Treatment with nocodazole had little if any effect on the mobility of early endosomes that only occupied trapped states (S1 Fig), indicating that microtubules play little if any role in active diffusion of early endosomes. Treatment of cells with latrunculin A reduced the mobility of early endosomes that only occupied trapped states, as shown by analysis of MSD vs. τ (Fig 9A), which contrasts with results with VSV

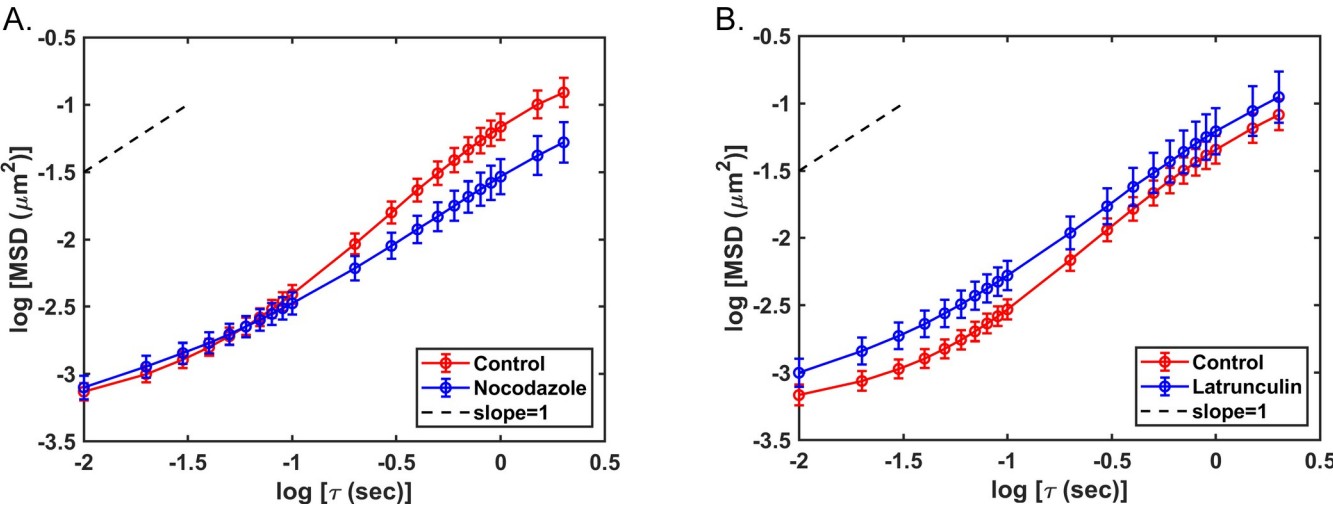

**Fig 7. τ analysis of early endosomes occupying directed states in control cells and cells treated with nocodazole or latrunculin A. MSD vs.** A549 cells transduced with a baculovirus vector that expresses the early endosome marker Rab5a fused to red fluorescent protein were not treated or treated with 15–20 μM nocodazole or 0.25 μM latrunculin A. Cells were imaged by fluorescence microscopy 15 to 30 min after addition of cellular inhibitor at 100 frames/s for 400 frames. Tracks of individual early endosomes occupying trapped states were analyzed at subpixel resolution with Video Spot Tracker software. Mean squared displacements (MSDs) were determined as a function of time delay (τ) and are shown as log-log plots for early endosomes occupying directed states in control and nocodazole-treated cells (A) and early endosomes occupying directed states in control and latrunculin A-treated cells (B). The dashed line has a slope of 1.0 (Brownian motion). The data shown are means ± 95% confidence intervals (for panel A, n = 149 tracks from control cells and 61 from nocodazole-treated cells; for panel B, n = 133 tracks from control cells and 67 from latrunculin A-treated cells) obtained in two sets of three independent experiments.

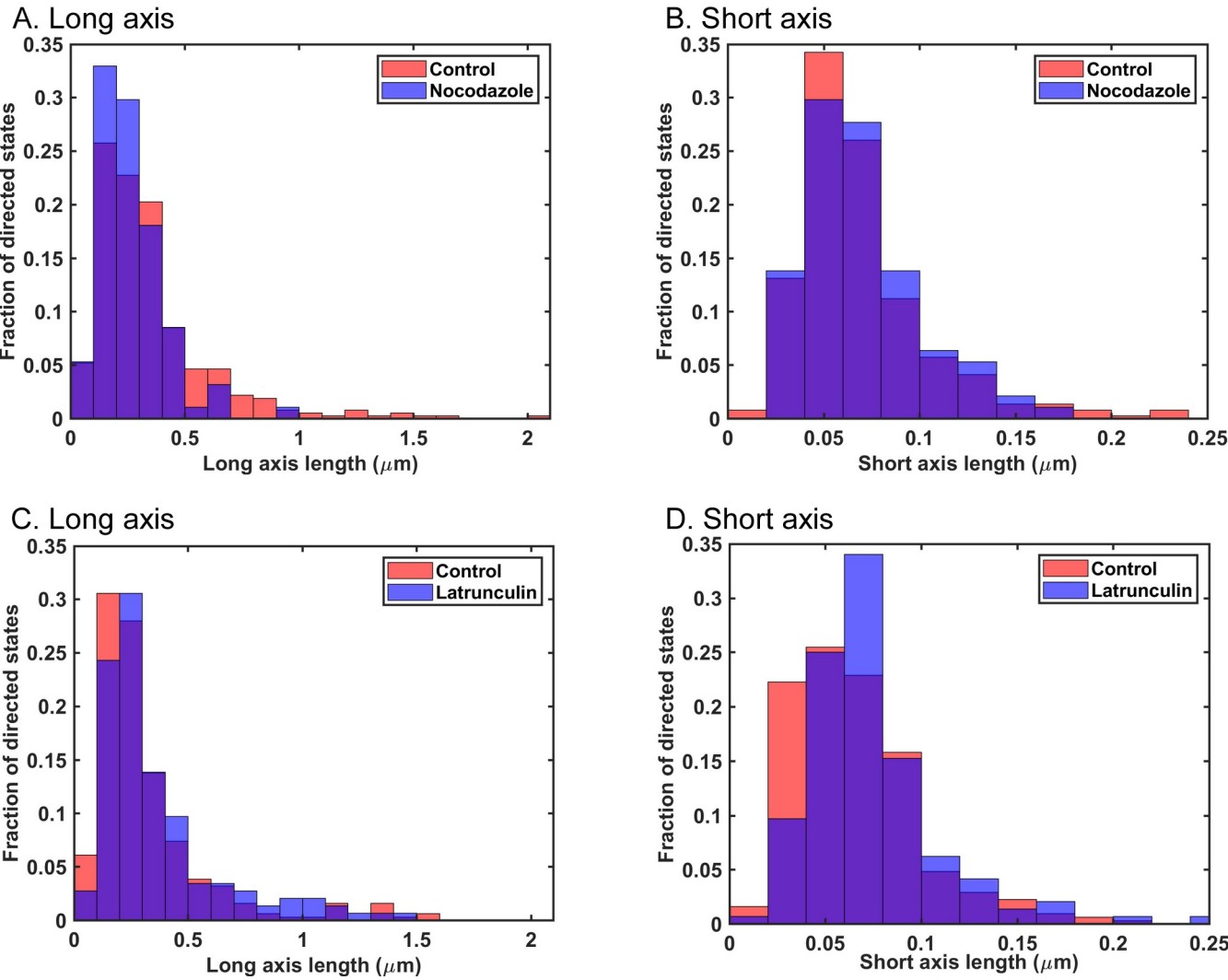

**Fig 8. Variational Bayesian analysis of sigma major and sigma minor for early endosomes occupying directed states in control cells and cells treated with nocodazole or latrunculin A.** Variational Bayesian analysis was used to calculate sigma major (A) and sigma minor (B) for early endosomes occupying directed states in control and nocodazole-treated cells, and sigma major (C) and sigma minor (D) for early endosomes occupying directed states in control and latrunculin A-treated cells. The state widths shown were calculated as the long axis (sigma major) and short axis (sigma minor) multiplied by 4 to account for the estimates of state widths as ± 2 sigma. Sigma major for nocodazole-treated cells (n = 94) was significantly smaller than sigma major in control cells (n = 365) (p < 0.05, Mann-Whitney test). Sigma minor for nocodazole-treated cells (n = 94) was slightly larger than sigma minor in control cells (n = 365), but the difference was not statistically significant (p > 0.05, Mann-Whitney test). Sigma major for latrunculin A-treated cells (n = 146) was significantly larger than sigma major in control cells (n = 311) (p < 0.05, Mann-Whitney test). Sigma minor in latrunculin A-treated cells (n = 146) was significantly larger than sigma minor in control cells (n = 311) (p < 0.05, Mann-Whitney test).

RNPs [11] and inclusion bodies (compare Fig 2C, where latrunculin treatment leads to increased mobility, to Fig 9A, where it decreased mobility). Fig 9B–9D show the results of Bayesian analysis of state width, dwell time, and distance between states for early endosomes occupying trapped states in control and latrunculin A-treated cells. There was no significant difference in the widths of the trapped states in control versus latrunculin A-treated cells [median state width = 0.093 μm in control and 0.092 μm in latrunculin A-treated cells (Fig 9B)]. This result indicates that actin filaments do not play a significant role in trapping early endosomes, which contrasts with the results with VSV RNPs [11] and inclusion bodies (Fig 4A). This result also indicates that changes in state width cannot account for the decreased mobility of these endosomes in latrunculin A-treated cells.

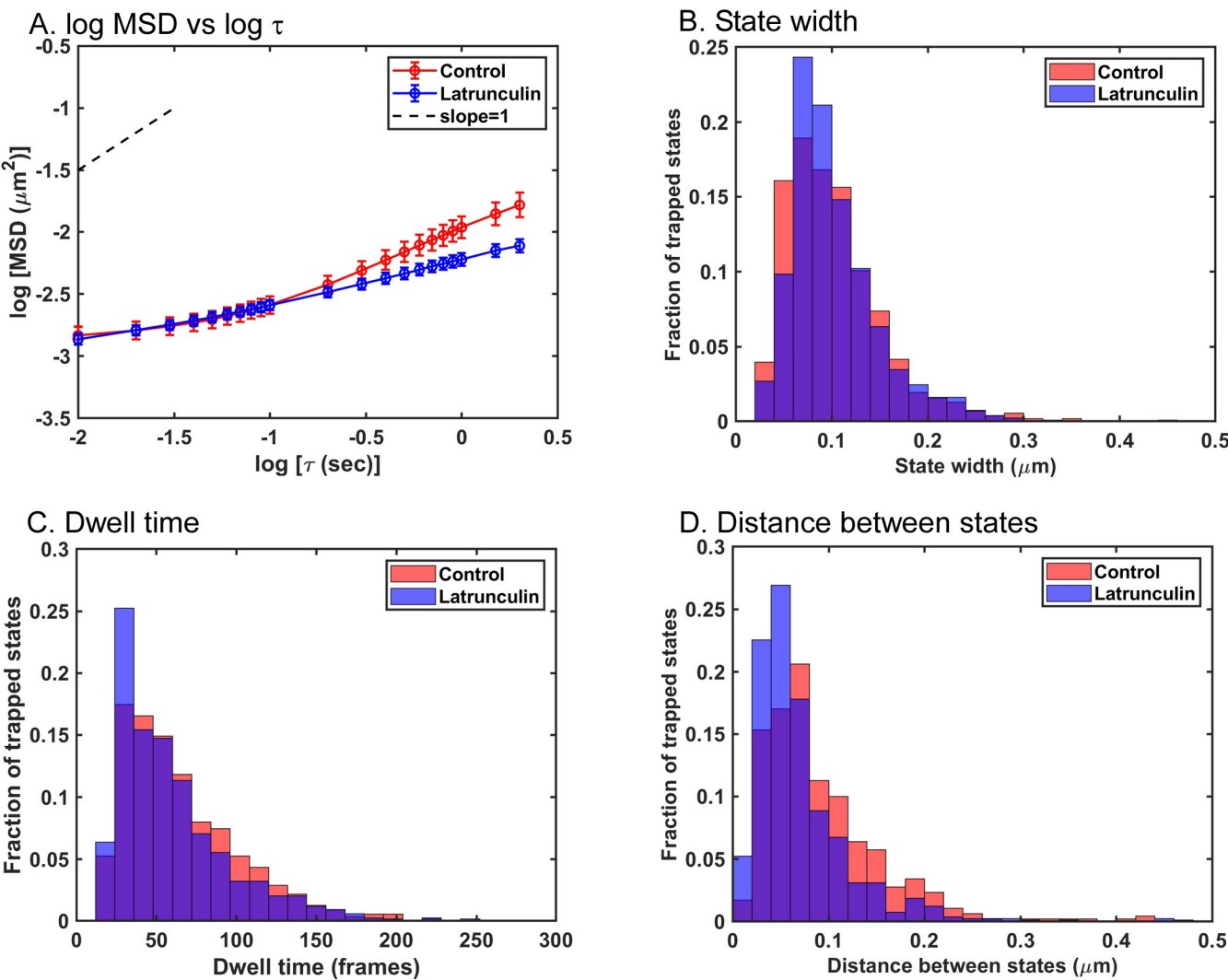

**Fig 9. Analysis of the movement of early endosomes occupying only trapped states in control cells and cells treated with latrunculin A.** Mean squared displacements (MSDs) were determined of early endosomes that only occupied trapped states as a function of time delay ($\tau$) and are shown as a log-log plot (A). The data shown are means ± 95% confidence intervals (n = 141 tracks from control cells and 262 from latrunculin A-treated cells) obtained in three independent experiments. Variational Bayesian analysis was used to calculate the widths of trapped states (B), dwell times (C), and the distance between states (D) of early endosomes that only occupied trapped states. States in latrunculin A-treated cells (n = 1292) were slightly smaller than those in control cells (n = 1082), but the difference was not statistically significant (p > 0.05, Mann-Whitney test). The dwell times of early endosomes occupying trapped states in latrunculin A-treated cells (n = 1173) were significantly shorter than those in control cells (n = 550) (p < 0.05, Mann-Whitney test). The distance between states was shorter for early endosomes occupying trapped states in latrunculin A-treated cells (n = 802 distances) than the distance between states in control cells (n = 470 distances) (p < 0.05, Mann-Whitney test).

The dwell times in trapped states decreased somewhat in latrunculin A-treated cells relative to control cells (Fig 9C), and the difference was statistically significant (p < 0.05, Mann-Whitney test). However, this cannot account for the decreased mobility, since shorter dwell times would lead to increased mobility following latrunculin A treatment. Instead, the decreased mobility of early endosomes occupying trapped states following latrunculin A treatment was due to decreased distance between trapped states [median distance between states = 0.07 μm in control cells, and 0.05 μm in latrunculin A-treated cells (Fig 9D), which was statistically significant (p < 0.05, Mann-Whitney test)]. This result indicates that actin filaments play a role in the movement of early endosomes from one state to the next. Furthermore, like VSV

inclusion bodies, the distance between states was shorter than the size of the particle, further supporting the hypothesis from results with VSV inclusion bodies that state-to-state movement is due to trap movement, not the particle jumping from trap to trap.

## Discussion

Previous research from our group demonstrated that active diffusion of VSV RNPs consists of bouncing back and forth within traps composed in part of actin filaments and hopping from trap to trap facilitated by myosin II motors [10, 11]. This raised the question of whether the same phenomena occurred with larger particles of different types. This was addressed in the present study by analyzing the motion of VSV inclusion bodies as examples of membraneless cellular compartments and early endosomes as examples of cellular membranous organelles (Figs 3, 5 and 6). The general pattern of movement of both VSV inclusion bodies and early endosomes was similar to that of VSV RNPs, consisting of periods of motion in trapped states, and occasional movement to different locations, where confined motion continued. However, the cytoskeletal elements involved in trapping were different. For VSV inclusion bodies, actin filament depolymerization by latrunculin A treatment increased mobility by increasing the width of the trapped states (Figs 2 and 3). Thus, like individual VSV RNPs, actin filaments likely form at least part of the traps that confine motion of inclusion bodies.

For cellular early endosomes occupying trapped states, neither treatment with latrunculin A nor disruption of microtubules by nocodazole treatment affected the width of trapped states (Fig 9B and S1B Fig). These data show that neither actin filaments nor microtubules are major components of traps for early endosomes. This suggests that other cytoskeletal elements like intermediate filaments serve as major trap components for early endosomes. Distances between cytoplasmic intermediate filaments visualized by electron microscopy are in the range from ~0.2 to ~1 µm [30–32], similar to the sizes of the early endosomes analyzed here (Table 1). Furthermore, cells from transgenic mice that lack vimentin intermediate filaments have increased mobility of cytoplasmic vesicles, suggesting that intermediate filaments play a role in trapping cellular organelles in this size range [33]. Treatment with latrunculin A did decrease the distance between states of early endosomes (Fig 9D), which is consistent with actin filaments playing a role in moving from state to state. This could be due to actin filaments that interact directly with early endosomes or indirectly through interactions between actin filaments and intermediate filaments [34].

A key result of the present study is that for both VSV inclusion bodies and early endosomes, the extents of confined motion (i.e. Bayesian states) were much smaller than the particles themselves (Figs 4A and 9B), indicating that nearly all of the trap is occupied by the particle itself, since the actual size of a trap that confines motion is measured by the size of the particle plus the extent of its movement. In our previous studies of individual VSV RNPs [10, 11], the size of the particles was smaller than the extent of motion within their traps, and the sizes of the traps were estimated only by the limits of the motion of the particles' center of mass. Trap diameters measured this way ranged from 0.05 to 1 µm with a median of 0.26 µm. It was acknowledged that the actual trap size would be slightly larger depending on how close the center of mass of the RNP is able to approach the edge of the trap [11], which is a complicated question because of factors such as time-dependent fluctuations in the shape of the flexible VSV RNP [35].

Another key result is that the distance between trapped states was much smaller than the size of the particles themselves (Figs 4C and 9D). This result suggests that state to state movement is due to movement between the states making up a single trap, not the particle hopping from one trap to another. In the previous study of VSV RNPs, analysis of the time dependence

of the particles' mean squared displacements showed that the traps are dynamic, expanding and contracting several times per second due to the action of myosin II motors [11]. Trap expansion and contraction could be due to two or more individual actin filaments that form the trap moving in opposite directions relative to each other without markedly changing the position of the center of the trap. However, movement of two or more actin filaments in the same direction (or movement of one filament) would lead to movement of the particle and the center of its trap to a new position, apparent as a different Bayes state. Thus, it is likely that actin filaments not only act as components of traps, but also help large particles move from state to state.

Although the focus of this study was on random motion, the variational Bayesian analysis approach can also be used to analyze directed motion [10]. The properties of the ellipse that define a directed state provide information about the distance traveled by a particle in a directed state from the length of its long axis and the extent of random motion within a directed state from the length of its short axis (Fig 5). VSV RNPs and inclusion bodies undergo very little directed motion in the cytoplasm [[10, 11]; data not shown], but a substantial percentage of early endosomes undergo directed motion (Fig 6E). Treatment with either latrunculin A or nocodazole decreased the number of directed states of early endosomes (Fig 6E), consistent with previous data that early endosomes travel on both microtubules and actin filaments for directed motion in the cytoplasm [21–23]. Analysis of the residual directed motion following inhibitor treatment suggested that microtubules are responsible for more of the directed state mobility of early endosomes than actin filaments (Fig 7), and that early endosomes travel shorter distances on actin filaments than on microtubules as shown by changes in the lengths of the long axes of the ellipses derived from the Bayesian analysis (Fig 8A and 8C). Analysis of changes in the short axes showed that actin filaments keep early endosomes within their directed states when traveling along microtubules (Fig 8D).

Two aspects of our approach have made it particularly well suited to investigating the mechanisms of active diffusion. First, rapid data acquisition captured the rapid bouncing back and forth of particles within their traps. This motion of individual VSV RNPs is well described by thermal (Brownian) motion confined within a spring-like (harmonic) potential well, such as a trap with flexible walls or a flexible tether [10]. In most cases we have interpreted this motion as indicating traps rather than tethers because of the well-established role of the cytoskeleton in trapping model particles in cell-free studies of cytoskeletal elements containing a limited number of components [7–9]. However, early endosomes undergoing directed motion are clearly tethered to either microtubules or actin filaments. These flexible tethers give rise to confined motion in random directions reflected in the short axes of the ellipses derived from the variational Bayesian analysis (Fig 5B).

The second aspect of our approach that provided new insight into the mechanisms of active diffusion was the variational Bayesian analysis of particle tracks, in which the motion within a trap was modeled as a 2D Gaussian distribution of data points, and Bayes' theorem was used to determine the most probable number of states occupied by the particle. Although this approach effectively models the behavior of particles undergoing active diffusion, it does have limitations. One limitation is that the traps are modeled as static structures, when we know from the time dependence of particles' mean squared displacements that the traps are dynamic, expanding and contracting several times per second [11]. Thus, what appear to be two or more traps may represent the same trap at different points in time. Indeed, this is the interpretation of the results of the experiments presented here in which the distances between Bayes states are too short to be due to the particles hopping from one trap to another (Figs 4A and 9B). Another limitation of the approach is that the image data are limited to two dimensions ($x$ and $y$). This leaves open the question of whether active diffusion in the third dimension ($z$) has the same characteristics. This is a biologically interesting question particularly for

cells with a flattened morphology, like A549 cells, in which the structure of the cytoskeleton in the $z$ dimension may be quite different. The time resolution in the $z$ dimension of current instrumentation is not as high as that in the $x$-$y$ plane, but future developments in technology and analytical methods are likely to be able to address this question.

Finally, the results presented here lead to consideration of conditions under which a particle will undergo active diffusion by hopping from one trap to another versus the movement of the trap instead. The size and flexibility of the particle are clearly major determinants. Distances between cytoplasmic actin filaments visualized by electron microscopy range from around 50 nm in actin-rich areas to >200 nm in areas less actin-rich [4], which is the same as the size range of the traps that confine the motion of individual VSV RNPs [10, 11]. Particles at one extreme of the size range (<50 nm) are relatively free to move by Brownian diffusion with little or no interference by actin filaments or other elements that would trap larger particles. Thus, nearly all their motion would consist of diffusing from trap to trap, with little if any effect of movement of actin filaments. At the other extreme of the size range (e.g., >500 nm like the particles analyzed here), these particles would not be able to fit through the spaces between actin filaments or other cellular elements like intermediate filaments and would depend solely on movement of the traps for active diffusion. Particles of intermediate size, such as individual VSV RNPs, may undergo a mixture of trap movement and hopping between traps. Thus, mechanisms of active diffusion can be viewed as a continuum dependent on the size and properties of the particle, with varying proportions of hopping from trap to trap versus motion dependent on trap dynamics.

## Materials and methods

### Cell lines

A549 human lung adenocarcinoma cells (CVCL_0023; CCL-185; American Type Culture Collection [ATCC]) and BHK-21 (C-13) baby hamster kidney cells (CVCL_1915; CCL-10; [ATCC]) were cultured in Dulbecco's modified Eagle medium as described previously [11]. Cell authentication was validated through short tandem repeat (STR) analysis performed by the ATCC. Mycoplasma contamination was not detected in either cell line as verified by the ATCC and by the Cell Engineering Shared Resource of the Comprehensive Cancer Center of Wake Forest University.

### Recombinant virus

Recombinant VSV containing enhanced green fluorescent protein (eGFP) inserted into the hinge region of the P protein (VSV-PeGFP) was acquired from Dr. Asit Pattnaik (University of Nebraska-Lincoln) and derived from a cDNA clone whose sequence was determined by Das et al. [26]. Stocks of VSV-PeGFP were prepared by infection of BHK-21 cells as described [11].

### Transduction with CellLight™ Early Endosomes-RFP, BacMam 2.0

A549 cells were transduced with CellLight™ Early Endosomes-RFP, BacMam 2.0 (C10587; Thermo Fisher Scientific) to fluorescently label early endosomes with an RFP tag. Cells were seeded in 35 x 10 mm glass-bottom dishes (70670–02; Fisher Scientific) at $1 \times 10^5$ cells/dish. After overnight incubation at 37˚C and 5% $CO_2$, cells were transduced for 2–4 hours at 30 particles per cell (PPC) with constant, gentle shaking. Cells were then treated with a 1:1000 dilution of BacMam enhancer solution from a BacMam Enhancer Kit (B10107; Thermo Fisher Scientific) for 1.5 hours. Cells then had their media refreshed and were incubated at 37˚C and 5% $CO_2$ for > 16 hours prior to imaging.

## Cellular inhibitors

Latrunculin A (593 μM; 10010630; Cayman Chemical Company) was purchased prepared in 100% ethanol. Nocodazole (25 mM; 74072; StemCell Technologies) was prepared in dimethyl sulfoxide (DMSO). Infected cells were treated with latrunculin A (0.25 μM), and transduced cells were treated with latrunculin A (0.25 μM) or nocodazole (17 μM for the first experiment, 20 μM for subsequent experiments) for the times indicated prior to imaging.

## Live-cell fluorescence imaging

A549 cells were seeded in 35 x 10 mm glass-bottom dishes (70670–02; Fisher Scientific) at 1 x $10^5$ cells/dish and incubated overnight at 37˚C and 5% $CO_2$. After 24 hours, cells were infected with VSV-PeGFP at an MOI of 3. After a one-hour adsorption period at 37˚C, the inoculum was removed, and cells were refed with FluoroBrite™ DMEM (A1896701; Thermo Fisher Scientific) containing 2% heat-inactivated fetal bovine serum. Cells transduced with CellLight™ Early Endosomes-RFP reagent were also refed with FluoroBrite™ DMEM prior to imaging. Cells were imaged using a Nikon Ti inverted epifluorescence microscope fitted with a 60X NA 1.4 oil-immersion objective. The microscope was set up on an optical breadboard (Newport) mounted onto four pneumatic vibration isolators (Thorlabs PWA075), and the stage environment was set to 37˚C and 5% $CO_2$. Light-emitting diodes served as the fluorescence light sources for GFP and RFP (Lumencor Spectra III). A Hamamatsu ORCA-Fusion scientific complementary metal oxide semiconductor (sCMOS) camera (C1440-20UP; Melrose, New York) with 6.5 x 6.5 μm pixels and a CoaXPress (CL) digitized the images as 16-bit tif stacks and sent the stacks to a computer. Images were captured at 100 frames/s for 4 s at a resolution of 9.23 pixels/μm. The camera software (HCImage) generated TTL pulses which synchronized the fluorescence light source with image acquisition. This exposed cells to excitation light only during image acquisition to minimize bleaching.

## VSV inclusion body and early endosome size measurements

The sizes of inclusion bodies were quantified using the freehand tools in ImageJ [Fiji Version 1.8.0_172; [36]]. Single frame images were created from previously collected 16-bit tif stacks, and the minimum fluorescence intensity threshold was increased above the intensity of the individual VSV RNPs in the cell to identify inclusion bodies with intensities above the threshold. Because the inclusion bodies were irregular in shape, their sizes were determined as the area defined by the number of pixels with intensity above the threshold. The magnifying glass tool was used to zoom in to the pixel level on randomly selected inclusion bodies within the single frame images. The Rectangle tool was used to crop the image to include the pixels with intensity above the threshold, and the cropped image was saved as a text image. The number of pixels with intensities above the threshold and the resulting area were calculated using MATLAB (version 9.9.0.1570001; Mathworks, Inc.). The sizes (n = 97) included in Table 1 are the approximate diameters calculated as the diameter of a circle with equal area.

The sizes of early endosomes were quantified using the Plot Profile and freehand tools in ImageJ. Single frames were selected from previously collected 16-bit tif stacks, and the magnifying glass tool was used to zoom in to the pixel level on randomly selected early endosomes. The line tool was used to create a line that passed through the pixel with maximum intensity. The fluorescence intensity was quantified using the Plot Profile function. The line tool and Plot Profile function were then used to quantify the nearby background of the same fluorescent early endosome. These data were saved as Excel files, the background was subtracted, and the intensities were fit to a normal distribution in MATLAB to calculate diameter as 2 x sigma [37]. The mean diameter and standard deviation were also calculated using MATLAB. The

sizes in Table 1 were determined from 50 early endosomes per experiment in four independent experiments (n = 200) from 25 control cells total.

### VSV inclusion body and early endosome tracking

Images of individual cells were cropped using ImageJ from the tif stacks obtained in live-cell fluorescence imaging experiments. The cropped files were analyzed by Video Spot Tracker (VST) software (version 8.01; Computer Integrated Systems for Microscopy and Manipulation, University of North Carolina-Chapel Hill [CISSM-UNCCH]) to track VSV inclusion bodies and early endosomes. Both were tracked at subpixel resolution using the VST cone function, a precision value set to 0.001, sample spacing set to 0.2, and radius set to 5. At these settings, the localization error within the positions of similarly sized (1 µm) fluorescent latex beads stuck to the surface of a cover slip was 0.023 µm.

For images containing VSV inclusion bodies, the minimum fluorescence intensity threshold was increased to "blanket-out" individual VSV RNPs present in the cell. The remaining viral particles were chosen for tracking based on the following properties: (i) fluorescence intensity above the minimum fluorescence intensity threshold, (ii) did not encounter another particle within the detection radius, and (iii) remained in the focal plane for the entire 400 frames. Early endosomes present throughout the cell cytoplasm were chosen for analysis based on the following properties: (i) fluorescence intensity consistent with a single early endosome, (ii) did not encounter another particle within the detection radius, and (iii) remained in the focal plane for the entire 400 frames. For both VSV inclusion bodies and early endosomes, the vrpn log files generated by VST were converted to MATLAB files using the vrpnLogToMatlab utility (version 05.00; CISSM-UNCCH). For VSV inclusion bodies, 1–2 inclusion body tracks were obtained for each cell, and 15 to 45 cells were analyzed for each condition (control or treated) in each experiment. For early endosomes, 5 to 30 early endosome tracks were obtained for each cell, and 5 to 10 cells were analyzed for each condition (control or treated) in each experiment. The number of independent experiments is indicated in the figure legends. Mean squared displacements (MSDs) for each track were calculated as a function of time delay ($\tau$) using MATLAB as described [10] with $\tau$ ranging from 0.01 to 2 s.

### Variational Bayesian analysis

The variational Bayesian analysis approach has been previously described in detail in [10] and [11], and is briefly summarized here. Individual tracks comprise an *x* and *y* coordinate for that particle in each of 400 frames. These data were fit to two-dimensional (2D) Gaussian distributions. Gaussian mixture models with 1 to 6 Gaussian states were sequentially tested to determine which model had the highest probability based on the data, and to determine which Gaussian state each *x-y* pair most probably belonged. This was accomplished by assigning initial parameters for the Gaussian states (amplitude, center *x*, center *y*, variance *x*, variance *y*), which were used to calculate an initial probability lower bound (*L*) based on Bayes' theorem as described [28, 38, 39]. A first-order hidden Markov model was added to describe transitions between states from frame-to-frame. The parameters for the Gaussian states and the probabilities of each data point belonging to each state were alternately optimized to improve *L*. Finally, each data point was assigned to a single state by identifying the sequence of states with the highest global probability. If points were in states of duration 5 frames or less, they were reassigned to their adjacent states.

### Statistical analysis

The numbers of independent experiments and parameters of the Gaussian states derived from the analysis are indicated in the figure legends. Differences between control and inhibitor-

treated cells were tested for statistical significance ($p < 0.05$) by Mann-Whitney tests using either MATLAB or Prism software.

## Fraction of directed states of early endosomes

Individual Gaussian states were classified as either trapped or directed from analysis of the tracks of early endosomes. Tracks were initially sorted into those with eccentricity value greater than 2.5 and average slope of log-log plot of MSD vs. τ greater than 0.5. An average slope value of 0.5 was chosen because the log-log plots of MSD vs. τ of directed states were curved with slopes near 0 for short values of τ and slopes $> 1$ at longer values of τ to average $>0.5$. The third criterion used to classify a state as directed [10] was travel in a mostly unidirectional fashion from one end to the other of the long axis of the ellipse defined by ± 2 sigma. Visual inspection of 100 states defined by the first two criteria indicated that this criterion was satisfied as well. Fraction of states classified as directed was calculated by dividing the number of directed states by the number of all states analyzed for each experiment. For early endosomes in latrunculin A-treated or nocodazole-treated cells, n = 3 experiments. Values for controls in these two series were combined, thus n = 6 experiments. The means, standard deviations, and 95% confidence intervals of the data were calculated using Excel. The bar plot in Fig 6E was created using MATLAB.

## Supporting information

**S1 Fig. Variational Bayesian analysis of state widths, dwell times, and distance between states for early endosomes occupying trapped states in control cells and cells treated with nocodazole.** Mean squared displacements (MSDs) of early endosomes that only occupied trapped states were determined as a function of time delay (τ) and are shown as a log-log plot (A). The data shown are means ± 95% confidence intervals (n = 151 tracks from control cells and 238 from nocodazole-treated cells) obtained in three independent experiments. Variational Bayesian analysis was used to calculate the widths of trapped states occupied by early endosomes (B), dwell times of the early endosomes in those trapped states (C), and the distance between states (D). States in nocodazole-treated cells (n = 1353) were slightly larger than those in control cells (n = 1206), but the difference was not statistically significant ($p > 0.05$, Mann-Whitney test). The dwell times of early endosomes occupying trapped states in nocodazole-treated cells (n = 1149) were significantly shorter than those in control cells (n = 660) ($p < 0.05$, Mann-Whitney test). The distance between states was slightly shorter for early endosomes occupying trapped states in nocodazole-treated cells (n = 237 tracks) than the distance between states in control cells (n = 150 tracks) ($p > 0.05$, Mann-Whitney test). (TIF)

**S1 File. Supplemental data.** Excel file of data used to generate Table 1 and Figs 2–9 and S1 Fig. (XLSX)

## Author Contributions

**Conceptualization:** Steven J. Moran, Jed C. Macosko, George Holzwarth, Douglas S. Lyles.

**Data curation:** Steven J. Moran, George Holzwarth, Douglas S. Lyles.

**Formal analysis:** Steven J. Moran, Douglas S. Lyles.

**Funding acquisition:** Steven J. Moran, Douglas S. Lyles.

**Investigation:** Steven J. Moran, Ryan Oglietti, Kathleen C. Smith, George Holzwarth, Douglas S. Lyles.

**Methodology:** Steven J. Moran, George Holzwarth, Douglas S. Lyles.

**Project administration:** George Holzwarth, Douglas S. Lyles.

**Resources:** George Holzwarth.

**Software:** George Holzwarth, Douglas S. Lyles.

**Supervision:** George Holzwarth, Douglas S. Lyles.

**Validation:** Steven J. Moran, Douglas S. Lyles.

**Visualization:** Steven J. Moran, Douglas S. Lyles.

**Writing – original draft:** Steven J. Moran.

**Writing – review & editing:** Steven J. Moran, Jed C. Macosko, George Holzwarth, Douglas S. Lyles.

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
