## [Decision Letter · Decision Letter 0]

17 Jan 2024

PONE-D-23-25885Mechanisms of active diffusion of vesicular stomatitis virus inclusion bodies and cellular early endosomes in the cytoplasm of mammalian cellsPLOS ONE

Dear Dr. Lyles,

Thank you for submitting your manuscript to PLOS ONE. After careful consideration, we feel that it has merit but does not fully meet PLOS ONE’s publication criteria as it currently stands. Therefore, we invite you to submit a revised version of the manuscript that addresses the points raised during the review process.

Dear Dr. Lyles,

Thank you for your patience while your manuscript "Mechanisms of active diffusion of vesicular stomatitis virus inclusion bodies and cellular early endosomes in the cytoplasm of mammalian cells" was under peer review at PLoS One. It has now been seen by our referees, whose expertise and comments you will find at the end of this email.

From their reports, you will see that they find your work of interest. Nevertheless, they have some minor comments you need to address before we can accept your article for publication. As you can see from their comments, reviewer 1 has some concerns about the significance of the observed movement for trapped inclusion bodies and the effect other phenomena could have on the results, and reviewer 2 has some comments about the statistics and methodology employed.

Again, thank you very much for your patience,

We look forward to receiving your revised manuscript.

Kind regards,

Gilberto Jose Betancor Quintana, Ph.D

Academic Editor

PLOS ONE

Journal Requirements:

Reviewers' comments:

Reviewer's Responses to Questions

**Comments to the Author**

1. Is the manuscript technically sound, and do the data support the conclusions?

Reviewer #1: Yes

Reviewer #2: Yes

2. Has the statistical analysis been performed appropriately and rigorously? 

Reviewer #1: Yes

Reviewer #2: Yes

3. Have the authors made all data underlying the findings in their manuscript fully available?

Reviewer #1: Yes

Reviewer #2: Yes

4. Is the manuscript presented in an intelligible fashion and written in standard English?

Reviewer #1: Yes

Reviewer #2: Yes

5. Review Comments to the Author

Reviewer #1: In a previous article, Moran et al. showed that active diffusion of vesicular stomatitis virus (VSV) ribonucleoproteins (RNPs) in the cytoplasm consists of hopping between traps and that actin filaments and myosin II motors are components of this hop-trap mechanism.

In this new manuscript, they analyze the random motion of larger structures such as VSV inclusion bodies (that are the major sites of viral RNA synthesis and are formed by liquid-liquid phase separation) and cellular early endosomes.

They show that like VSV RNPs, inclusion bodies and early endosomes moved from one trapped state to another, but their size preclude hopping between traps and the apparent state-to-state movement is mediated by essentially driven by trap movement.

Finally, they show that latruculin A-induced actin depolymerization increases VSV inclusion body mobility most probaly by increasing the size of the traps. In contrast, depolymerization of actin or microtubules does not affect the size of traps that confine early endosome mobility, suggestingthat intermediate filaments are the major trap components for these organelles.

Athough the result is rather expected, the work is well done and the approaches used to analyze the data can be useful to cell biologists in general.

Their conclusion that mechanisms of active diffusion can be viewed as a continuum dependent on the size and properties of the particle, with varying proportions of hopping from trap to trap versus motion dependent on trap dynamics is also of general interest.

However, I have two comments that I would like the authors to adress.

1) As mentioned by the authors, the movement of trapped inclusion bodies and trapped early endosomes are very small compared to their size, on the order of 100nm (i.e. ~5% of the size of an inclusion body and ~20% of the size of an endosome). On the one hand, this is below optical resolution; on the other, it is in the pixel size range. This raises some questions about the significance of the observation (although I agree that there is a significant difference after latrunculin A treatment at least in the case of inclusion bodies).

2) Finally, can the authors completely exclude that other phenomena are at work? For example, in the case of the rabies virus, the inclusion bodies are of smaller average size after treatment with cytochalasin D which induces their fragmentation. Also, if the composition of the inclusion bodies is affected due to drug treatment, this may affect their properties (viscosity, surface tension and consequently the dynamics of their surface).

Reviewer #2: Moran et al., "Mechanisms of active diffusion of vesicular stomatitis virus inclusion bodies and cellular early endosomes in the cytoplasm of mammalian cells," infer the mechanism of movement for VSV inclusion bodies and early endosomes, finding differences in how the particulates move between traps and depend on the cytoskeleton. There is broad interest in understanding how such large compartments move as they cannot rely on passive diffusion. This is particularly important for cells where materials must be transported over long distances (e.g., neurons). I enjoyed the background on the flux of molecules and particulates in the introduction, which was unusually clear, and I commend the authors. I have some suggestions for them on their description of Liquid-liquid phase separation, which I find historically and experimentally inaccurate, but these are mostly minor or simply removable. The authors describe how they image their tracts for MSDs and Beysian analysis particularly well compared to the standard in the field, and the Bayesian analysis provides additional insights that this reviewer finds valuable. Furthermore, the number of tracts and images is relatively high, which makes me confident broadly in the author's work despite these measurements being difficult. I find the work compelling, only have minor comments/suggestions, and feel the manuscript warrants publication.

Comments:

1) Can the authors comment on the size distribution statistics and how many VSV inclusion bodies they analyze in Figure 1? A histogram (probably given N~100), violin plot, or a box plot showing all the points and the mean/standard deviation would be appropriate here to properly understand the authors' confidence.

The authors should validate whether the particles they track are similar in size. The authors note that size contributes significantly to how particles move within the cytoplasm. Thus, it should be either validated to be similar or possible confounding variables should be discussed.

2) The requirements for LLPS given in lines 106-107 are misleading and incorrect for near diffraction limited objects such as VSV inclusion bodies challenging to validate for LLPS. For example, concerning the claims, (1) two dots coming together could be diffusion within the diffraction limit of particles many nanometers away from each other, and (2) surface curvature is nearly impossible to quantify for most condensates by any technique (as single molecule methods require many particles and lack the detail to see a surface and others such as SIM require deconvolution that is ill-poised for cellular systems that have many aberration and non-gaussian noise considerations that are often overlooked, perhaps the only method suited is STED but few have access to these techniques with the proper fluorophores.) Concerning (iii), most argue that lessened internal motion is a hallmark of LLPS. The best assay to validate phase separation is the saturation assay, albeit even that has considerable caveats. Broadly, the authors, who don't require LLPS for the impact of their paper, should lessen or remove these claims of rules and instead cite the previous papers and move on.

3) Similarly, but more minor. Most in the field prefer to avoid the prefix "liquid-liquid" as it can be misleading to many outside of the polymer field, where the term liquid tends to assume rapid dynamics where polymer liquids can be rather viscoelastic liquids ("glassy"). For example, the most well-accepted phase separation-driven condensate, the nucleolus, has recently been found to have complex viscoelastic properties (Riback et al. et al., Mol Cel 2023). Thus, the language should be corrected in line 102 and in that paragraph. While it is fair and consistent with Flory's use of the term liquid-liquid as a prefix to phase separation to describe polymeric viscoelastic solutions as being driven through those properties, it probably isn't advisable given the contentious nature of LLPS (Musacchio EMBO 2022).

4) Line 115, "like our previous study," is missing a citation.

6. PLOS authors have the option to publish the peer review history of their article (what does this mean?). If published, this will include your full peer review and any attached files.

Reviewer #1: No

Reviewer #2: No

---

## [Author Response · Author response to Decision Letter 0]

8 Feb 2024

Response to Reviewers’ Comments 

We thank the reviewers for their thoughtful comments and have revised the manuscript in response as described here. Changes from the original version are indicated in red font in the marked revised version. 

Reviewer 1:

1) “The movement[s] of trapped inclusion bodies and trapped early endosomes are very small . . . on the order of 100 nm . . . below optical resolution. This raises some questions about the significance of the observation.” 

Response: The Video Spot Tracker software uses data from multiple pixels to calculate the position of the center of the fluorescent particles to subpixel resolution, but the reviewer raises an important point. Localization error, which can arise from stage vibrations, and the subpixel tracking error generated by Video Spot Tracker, was determined from the localization error in the position of 1μm latex beads stuck to the surface of a cover slip to be 0.023 μm. Thus although the distances traveled from trap to trap are small relative to the size of the particles, they are at least 3- to 4-fold above the localization error. These points are included in lines 227-230 and 523-525 in the revised manuscript.

2)“Can the authors completely exclude that . . . the inclusion bodies are of smaller average size after treatment . . . [or] if the composition of the inclusion bodies is affected due to drug treatment.”

Response: The sizes of the inclusion bodies that were analyzed in latrunculin A-treated cells were not significantly different from those in control cells. Histograms illustrating this point are included in the revised manuscript as Fig 1E, and the statistical results are described in lines 168-170 and in the legend for Fig 1E. Much more difficult to exclude is the possibility that drug treatment changes the composition of the inclusion bodies. This possible explanation for the results is included in lines 209-211.

Reviewer 2:

1) “Can the authors comment on the size distribution statistics and how many VSV inclusion bodies they analyze in Figure 1? A histogram . . . would be appropriate here. . . . The authors should validate whether the particles they track are similar in size.” 

Response: This is similar to comment 2 of Reviewer 1. The addition of the histograms in Fig 1E and the description of the statistics in the revised manuscript are in response to both reviewers’ comments.

2)“The requirements for LLPS given in [the original manuscript] are misleading and incorrect. . . .Broadly the authors, who don’t require LLPS for the impact of their paper, should lessen or remove these claims of rules and instead cite the previous papers and move on.”

Response: This was an excellent suggestion, especially in light of comment 3. The indicated text was deleted in the revised manuscript.

3)“Most in the field prefer to avoid the prefix "liquid-liquid" . . . it probably isn't advisable given the contentious nature of LLPS (Musacchio EMBO 2022).”

Response: The reviewer’s and Professor Musacchio’s points are well-taken, and many claims of liquid-liquid phase separations (LLPS) could be due to alternate mechanisms, such as rapidly exchanging site-specific interactions. We have changed references to LLPS in the revised manuscript to the more general term “membraneless cellular compartments” as in the title of the referenced paper, which is also now cited in the revised text of the manuscript (lines 99-104).

4)“Line 115, "like our previous study," is missing a citation.” 

Response: Thank you, citations added (now line 110).

---

## [Editor Report · Decision Letter 1]

13 Feb 2024

Mechanisms of active diffusion of vesicular stomatitis virus inclusion bodies and cellular early endosomes in the cytoplasm of mammalian cells

PONE-D-23-25885R1

Dear Dr. Lyles,

We’re pleased to inform you that your manuscript has been judged scientifically suitable for publication and will be formally accepted for publication once it meets all outstanding technical requirements.

Kind regards,

Gilberto Jose Betancor Quintana, Ph.D

Academic Editor

PLOS ONE

Additional Editor Comments (optional):

Dear Prof Lyles,

Thank you very much for the revised version of your manuscript,

I am glad to announce that your revised manuscript fulfil all the reviewers comments and therefore will be accepted for publication at PLoS One,

Once again, I would like to thank you for your patience during the revision process,

Best regards,

Gilberto
---

## [Editor Report · Acceptance letter]

4 Mar 2024

PONE-D-23-25885R1 

PLOS ONE

Dear Dr. Lyles, 

I'm pleased to inform you that your manuscript has been deemed suitable for publication in PLOS ONE. Congratulations! Your manuscript is now being handed over to our production team.

Kind regards, 

on behalf of

Dr. Gilberto Jose Betancor Quintana 

Academic Editor

PLOS ONE